# Long-Term Survival and Regeneration Following Transplantation of 3D-Printed Biodegradable PCL Tracheal Grafts in Large-Scale Porcine Models

**DOI:** 10.3390/bioengineering11080832

**Published:** 2024-08-14

**Authors:** Sen-Ei Shai, Yi-Ling Lai, Yi-Wen Hung, Chi-Wei Hsieh, Kuo-Chih Su, Chun-Hsiang Wang, Te-Hsin Chao, Yung-Tsung Chiu, Chia-Ching Wu, Shih-Chieh Hung

**Affiliations:** 1Department of Thoracic Surgery, Taichung Veterans General Hospital, Taichung 407219, Taiwan; windjay77@hotmail.com; 2Department of Applied Chemistry, National Chi Nan University, Nantou 545301, Taiwan; 3Institute of Clinical Medicine, National Yang-Ming Chiao-Tung University, Taipei 112304, Taiwan; 4Animal Radiation Therapy Research Center, Central Taiwan University of Science and Technology, Taichung 406053, Taiwan; hongiw@yahoo.com.tw; 5Terry Fox Cancer Research Laboratory, Translational Medicine Research Center, China Medical University Hospital, Taichung 404327, Taiwan; 6School of Medicine, National Cheng Kung University, Tainan 701401, Taiwan; peter100yahoo@gmail.com (C.-W.H.); joshccwu@mail.ncku.edu.tw (C.-C.W.); 7Department of Medical Research, Three Dimensional Printing Research and Development Group, Taichung Veterans General Hospital, Taichung 407219, Taiwan; kaoche2000@gmail.com (K.-C.S.); wangch@vghtc.gov.tw (C.-H.W.); 8Division of Colon and Rectal Surgery, Department of Surgery, Chiayi and Wangiao Branch, Taichung Veterans General Hospital, Chiayi 600573, Taiwan; thchao@vghtc.gov.tw; 9Department of Medical Research and Education, Taichung Veterans General Hospital, Taichung 407219, Taiwan; ytchiu@vghtc.gov.tw; 10Department of Cell Biology and Anatomy, College of Medicine, National Cheng Kung University, Tainan 701401, Taiwan; 11Integrative Stem Cell Center, China Medical University Hospital, Taichung 404327, Taiwan; hung3340@gmail.com; 12Institute of New Drug Development, China Medical University, Taichung 404328, Taiwan

**Keywords:** 3D-printed tracheal graft, polycaprolactone (PCL), long-term survival, large-scale porcine

## Abstract

Polycaprolactone (PCL) implants in large animals show great promise for tracheal transplantation. However, the longest survival time achieved to date is only about three weeks. To meet clinical application standards, it is essential to extend the survival time and ensure the complete integration and functionality of the implant. Our study investigates the use of three-dimensional (3D)-printed, biodegradable, PCL-based tracheal grafts for large-scale porcine tracheal transplantation, assessing the feasibility and early structural integrity crucial for long-term survival experiments. A biodegradable PCL tracheal graft was fabricated using a BIOX bioprinter and transplanted into large-scale porcine models. The grafts, measuring 20 × 20 × 1.5 mm, were implanted following a 2 cm circumferential resection of the porcine trachea. The experiment design was traditionally implanted in eight porcines to replace four-ring tracheal segments, only two of which survived more than three months. Data were collected on the graft construction and clinical outcomes. The 3D-printed biosynthetic grafts replicated the native organ with high fidelity. The implantations were successful, without immediate complications. At two weeks, bronchoscopy revealed significant granulation tissue around the anastomosis, which was managed with laser ablation. The presence of neocartilage, neoglands, and partial epithelialization near the anastomosis was verified in the final pathology findings. Our study demonstrates in situ regenerative tissue growth with intact cartilage following transplantation, marked by neotissue formation on the graft’s exterior. The 90-day survival milestone was achieved due to innovative surgical strategies, reinforced with strap muscle attached to the distal trachea. Further improvements in graft design and granulation tissue management are essential to optimize outcomes.

## 1. Introduction

In the realm of tracheal reconstruction, addressing the intricate challenges posed by extensive tracheal resection has necessitated the exploration of diverse reconstructive strategies, including the use of autografts, allografts, and prosthetics [1,2,3,4]. Despite these efforts, achieving long-term success and functional regeneration of the trachea remains elusive [5]. This challenge has spurred interest in tissue engineering as a promising avenue for developing sustainable, tissue-engineered tracheas that meet the needs of patients undergoing long-segment tracheal resections [6]. The advent of 3D printing technology has revolutionized the field, offering unprecedented precision in creating biodegradable polycaprolactone (PCL) scaffolds tailored to individual anatomical specifications [7,8,9]. PCL, known for its mechanical strength and biocompatibility, stands out among biodegradable polymers for its suitability in 3D printing applications [10,11]. This eliminates the need for a time-consuming decellularization process [12,13] and harmful solvents [14], allowing for the fabrication of scaffolds that closely replicate the intricate architecture of the natural trachea. Such advances have not only paved the way for structural repairs but also promise functional restoration over prolonged periods, marking a significant leap forward from traditional methods that provided only temporary solutions [6,15].

The utilization of large animal models plays a crucial role in this research, bridging the gap between small-scale experiments and human clinical applications [16]. While small-animal models have contributed valuable insights into the design and testing of bioengineered tracheas and grafts [17,18,19,20], their limitations become apparent when assessing graft functionality in vivo, primarily due to the inability of small airways to accommodate bronchoscopic instrument [21]. Thus, large-scale animal models, with airway sizes more comparable to humans, are essential for a realistic evaluation of tracheal transplantation and long-term graft viability. The PCL implants in large animals show great promise for tracheal transplantation. However, the longest survival time achieved to date is only about three weeks. To meet clinical application standards, it is essential to extend the survival time and ensure the complete integration and functionality of the implant. Our current research extends beyond the scope of previous studies by providing a comprehensive analysis of the entire neotissue layer in situ, including cartilage, muscle, adipose, and glandular tissues derived from mesenchymal cells [22,23]. We have documented the chronological development of chondrogenesis, glandogenesis, myogenesis, and adipogenesis, elucidating the intricate processes of cellular development within the regenerative tissue. This lucid examination highlights the crucial role of airway remodeling and the formation of new tissue.

The process toward achieving a successful long segment of circumferential tracheal replacement has encountered numerous challenges, from the technical difficulties of integrating synthetic grafts to the ethical and logistical issues associated with donor-dependent methods [24,25,26,27,28]. The transition toward biosynthetic scaffolds represents a critical advancement, offering a viable platform for in vivo tissue engineering that circumvents the limitations of previous approaches. This strategy not only provides immediate structural support but also promotes the emergence of new tissue, laying the groundwork for sustained graft survival and integration into the body’s ecosystem [29].

By harnessing the capabilities of 3D printing to develop PCL tracheal grafts tailored for large-scale porcine models, our research aims to surmount the longstanding obstacles faced in tracheal reconstruction. This initiative seeks to create a versatile platform for investigating the potential of customizable tracheal scaffolds, thereby enhancing not only the survival prospects but also the regeneration capabilities of tracheal tissues.

## 2. Materials and Methods

### 2.1. Design and Preparation of PCL Tracheal Grafts

The design and preparation process for PCL tracheal grafts (Appendix A) involved 3D printing using a CELLINK BIOX™ bioprinter (Appendix A). The specified graft dimensions incorporated six strategically placed distribution holes, as outlined by G-code parameters. The fabrication commenced with ester-terminated PCL (CAS-n 24980-41-4, CELLINK, TP-60505) pellets, a biodegradable polyester, heated to a molten state, then extruded in layers to form the graft, requiring 3–4 h. To summarize, we used Cellink HeartWare 2.4 software to edit the G-code printing program. We produced the best graft by printing with both sterilized support material “CELLINK START” (CELLINK, IK-190000) and PCL in different cartridges. This method comprised alternating PCL and washable support materials for the inner and outer layers, with PCL serving as the filler material. The maximum flexibility of the implant was attained with 20 mm in length, 20 mm in diameter, a thickness of 1.5 mm and a 25% infill. This method created implants with fine and regular pores, resulting in optimal printing conditions. The final PCL grafts featured four deliberately placed perforations to promote uniform cell ingrowth and integration with the native trachea after implantation (Appendix A). These grafts were comparable in thickness to the trachea of a young pig, which is circular in contrast to the C-shape typical of a human trachea. The properties of the grafts were validated against those of a native porcine trachea using a JSV-H1000 system. After production, the grafts underwent refinement and sterilization with ethanol and UV light according to Kim et al., 2019 [30], preparing them for transplantation in large-scale animal models.

### 2.2. Methods Design for In Vivo Experiments

Eight LY female porcine were obtained from Pinguan Modern Livestock Co., Ltd., (Taichung, Taiwan). These animals were approximately three months old and weighed between 30 and 62.5 kg (Figure 1). The Taichung Veterans General Hospital Institutional Experimentation Committee provided ethical approval for the experimental procedures (Animal Welfare Protocol Number La-1111859, approved on 3 November 2022). Our study team had extensive experience in conducting large-scale animal trials involving tracheal transplantation, with over 30 tests utilizing various materials for 3D-printed grafts. In this study, the implants were made from aseptic 3D-printed grafts that were well-preserved. There was a surgical fixation method: porcine were lastly handled with Traction sutures of proximal and distal ends of trachea, tied together to alleviate tension at the anastomosis sites bilaterally (Appendix A), and subsequently reinforced with additional reinforcement in which dual-end needle 3-O prolene sutures were used to penetrate the strap muscle individually and tied over the anterior lateral surface of the muscle for tension relief of graft force for negative pressure of the thorax (Appendix A). The experimental procedure entailed resecting a 2 cm circumferential segment of the trachea, followed by an end-to-end anastomosis between the proximal and distal tracheal portions with non-absorbable 3-O prolene sutures. This approach closely resembled human tracheal surgery (Appendix A. Procedures of surgical intervention of PCL tracheal graft). Animal feeding and care techniques adhered to both national and international standards.

### 2.3. Anesthesia, Surgical Intervention and Postoperative Care

Each porcine was anesthetized with Zoletil 50 (8 mg/kg, IM, Virbac, Carros, France) and intubated using a 7.0 mm endotracheal tube. Anesthesia was maintained with 4% isoflurane. The animals were placed in a supine position, and their necks were shaved and sterilized with povidone iodine followed by 75% alcohol. A disposable sterile towel was used for draping. A cervical midline incision was made, and the strap muscles were separated to expose the upper trachea (Appendix A). A 2 cm circumferential segment of the trachea was excised and replaced with the 3D-printed PCL tracheal graft (Appendix A). Cross-table ventilation was used during the creation of the proximal end anastomosis, which was replicated at the distal end. The endotracheal tube was repositioned through the graft into the distal trachea after completing the proximal and posterior distal end anastomoses with a continuous running 3-O prolene suture. The anterior half of the distal anastomosis was completed with interrupted 4-O prolene sutures placed above the tube. Traction sutures of proximal and distal ends of the trachea were tied to alleviate tension at the anastomosis sites bilaterally (Appendix A). After confirming an airtight seal, the strap muscles were closed, a silicone Penrose drain was inserted, and the wound was sutured in layers. Postoperatively, the animals were extubated in the operating room and recovered without complications within 12 h, resuming normal activities and diet. Postoperative care included pain relief with diclofenac potassium (25 mg), antibiotic treatment with ampicillin (500 mg), and a mucolytic agent (bromhexine hydrochloride, 2 mg/mL), all administered with food under veterinary supervision.

### 2.4. Scheduled Postoperative Bronchoscopic Evaluation and Laser Ablation for Granulation Tissue

Post-transplant, the tracheal lumen was regularly evaluated using intraluminal telescopic visualization (bronchoscopy) with a 4.9 mm 0° endoscope (Olympus BF-260/CV-260/CLV-260, YUAN YU INDUSTRY Co., Ltd., Taipei, Taiwan), approximately on a weekly basis, to monitor graft integration and ensure airway patency (Appendix A). Sedation was maintained during these procedures. Stenosis was quantitatively assessed by comparing the diameters of stenosed and non-stenosed tracheal segments. When significant granulation tissue was observed at the anastomosis site, Argon-plasma Coagulation (FiAPC probe 1500 A/VIO200D, Soma Technology, Inc., Bloomfield, CT, USA) was utilized to ablate the tissue and maintain lumen patency. ERA Bioteq ENTERPRISE Co., Ltd. (Taipei, Taiwan) supplied the equipment for this process.

### 2.5. Tracheal Tissue Harvested and Gross Examination

Biodegradable PCL tracheal grafts, produced using a BIOX bioprinter, were implanted into large-scale porcine models with the goal of achieving a three-month long-term survival. Upon the study’s completion, the tracheal grafts and associated cylindrical neotissues were harvested. During the autopsies, these in situ regenerated tissues underwent gross examination, focusing on infection, necrosis, and the severity of tracheal stenosis at both proximal and distal anastomosis sites. The samples were then fixed in 10% formalin and embedded in paraffin for subsequent analysis (Appendix A).

### 2.6. Regenerated Tubular Tissue outside Graft by Histology Analysis

Chondrocyte markers, including Sox9, type II collagen, and aggrecan, were effectively stained. Proliferating cells were detected using the proliferating cell nuclear antigen (PCNA) assay. CD31 and αSMA, markers for blood vessels within the neocartilage, were employed to assess angiogenesis using the UltraVision LP Detection System HRP Polymer & DAB Plus Chromogen (Thermo, TL-060-HD, Thermo Fisher Scientific Inc., Waltham, MA USA). Photomicrographs were captured using NDP.view 2.6.8 software. The total pixel areas within the sections stained for Sox9, aggrecan, and PCNA were computed, taking into account positively stained chondrocytes in the immunohistochemistry (IHC) staining, with the assistance of Image J 2 software.

Tissue specimens were fixed in 10% formalin and then embedded in paraffin. Various histological stains were utilized to assess the regenerated tissues, including hematoxylin and eosin (H&E), alcian blue (ScyTek, ANC250, ScyTek, UT, USA), and safranin O/fast green (Cat. #8348a/Cat. #8348b). Images of the stained sections were captured using NDP.view 2.6.8 software. The number of chondrocytes, as indicated by the H&E-stained sections, was quantified using Image J software. The neocartilage matrix content, specifically glycosaminoglycans (GAGs), was determined from the alcian blue and safranin O/fast green-stained sections. The total pixel area within the cartilage sections stained by H&E was calculated, taking into account the distinct lacunar morphology. The data from the neocartilage and native tracheal cartilage of the test subjects were gathered and compared to that of a mature six-month-old porcine with a stable tracheal cartilage (Stage IV).

### 2.7. Statistical Analyses

Images from each slide underwent random review and analysis by an individual blinded to the research questions. The data are presented as mean ± SEM and were subjected to statistical comparison using the Kruskal–Wallis test or ANOVA, as appropriate. The number of chondrocytes across different tissue samples was compared utilizing the Dunn–Bonferroni post hoc test within the Kruskal–Wallis framework or the Bonferroni correction within ANOVA. Statistical significance was established at a *p*-value of less than 0.05.

## 3. Results

The implanted PCL tracheal grafts with dimensions of 20 mm in length, 20 mm in diameter, a thickness of 1.5 mm, and 25% infill (Appendix A). Table 1 and Figure 1 present a summary of the outcomes from transplanting the PCL tracheal grafts into a porcine model. The study utilized two methods of surgical procedures across eight female porcines that varied significantly in weight. Six exclusions were due to complications such as infection, necrosis, and dehiscence, and survived for less than three months due to respiratory failure. A notable improvement with two inclusions, recorded weight gain, and all animals surviving beyond three months, led to their elective sacrifice for the study.

### 3.1. Gross Specimen of Tissue Regeneration Following Long-Term Implantation into Large-Scale Porcines

To gain insights into the tracheal regeneration process, we collected and analyzed regenerated tissues 92 days after graft implantation into the trachea (Appendix A). The whole-mount tissue was divided into three sections, ranging from proximal to distal, with each region further subdivided into left and right parts (Figure 1 and Figure 2). Due to the inability to precisely localize the specimens without reference to the esophagus, the exact position of the regenerated tissue could not be determined. However, this division facilitated subsequent analysis. The figure is a composite of photographs and histology slides depicting neotissue growth at various regions (proximal, middle, and distal) around a tracheal graft in two different porcine subjects over a period of 90 days post-implantation. For porcine 1, which had a soft tissue infection: The images showed the gross morphology of the intact cylindrical neotissue at the proximal, middle, and distal regions, respectively, with the PCL tracheal rings visible inside the neotissue (Figure 2A,C,E). The corresponding histology slides, stained with Hematoxylin & Eosin (H&E), show red circles indicating areas of heterotopic ossification (HO) (Figure 2B,D,F). For Porcine 2, which did not have an infection, the images displayed the gross morphology of the neotissue, with only a half-ring of neotissue growth in image G compared to the intact rings in images I and K (Figure 2G,I,K). The histology slides for Pig2, again with red circles, highlight areas of HO (Figure 2H,J,L).

### 3.2. Tracheal Tissue Regeneration Followed by Tracheal Graft Implantation

This is a set of histological sections showing a comparison of regenerated tracheal tissues in a large-scale porcine model over 90 days, focusing on the proximal area of neotissue. It comprises two main images (Figure 3), A and E, each with a scale bar indicating 10 mm, which provide a comprehensive view of the regenerated tracheal tissues. Accompanying each main image are close-up views of specific areas of interest, marked by colored squares that correspond to insets B–D for image A and F–H for image E. Figure 3A shows the overall view of the regenerated tracheal tissue featuring integrated cartilage, abundant submucosal glands, and a smooth epithelium. Inset B (marked with a red square on image A) displays the intact neocartilage with signs of chondrogenesis at the tip. Inset C (marked with a green square on image A) highlights the abundant newly formed submucosal gland clusters. Inset D (marked with a yellow square on image A) presents a smoothly regenerated epithelium. Figure 3E provides an overall view of another sample of regenerated tracheal tissue, but with heterotopic ossification, irregular submucosal glands, and epithelium. Inset F (marked with a red square on image E) is a representative central area showing the lamellar bone trabeculae with intervening fibrous tissue that resembles fibrotic marrow. Inset G (marked with a green square on image E) illustrates irregular clusters of submucosal glands. Inset H (marked with a yellow square on image E) shows the epithelium in the tissue sample.

The series presents histological sections comparing the distribution and morphology of submucosal glands in the trachea of three-month-old and six-month-old porcine subjects, as well as in regenerated tracheal tissue after implantation into a large-scale porcine model for over 90 days (Appendix A). The submucosal glands from the native trachea of a three-month-old porcine appear to have a homogenous distribution throughout the tissue, characterized by uniform spacing and size (Appendix A). The submucosal glands of a six-month-old porcine’s native trachea show a heterogeneous distribution, with variable spacing and size, indicating a less uniform pattern compared to the three-month-old porcine trachea (Appendix A). The regenerated tracheal tissue, implanted in the porcine for over 90 days, features submucosal glands that are more abundant and larger in size than those in the native trachea (Appendix A).

### 3.3. The Impact of Bacterial Infection-Induced Inflammation after Tracheal Graft Transplantation and Heterotopic Ossification (HO) Observed 92 Days Post-Implantation

This analysis presents various histological features of tracheal neotissue in a large-scale porcine model 90 days after implantation: It showcases the overall histological structure, with a focus on areas of HO (Figure 4A). It reveals the presence of bacterial clumps, suggesting an area of microbial colonization (Figure 4B). It indicates congestion in the submucosal area, likely due to increased vascularity or inflammation (Figure 4C). It highlights a polyp formation within the neotissue (Figure 4D). It details the tissue morphology surrounding the HO (Figure 4E). Each colored square (yellow, red, blue, green) corresponds to magnified views at 100× within panels B–E, offering a closer look at the specific histological features identified in the neotissue.

### 3.4. Characteristics of Vascular Canals (VCs) in the Evolution of Chondrogenesis: Derived from Perichondrium as Chondro-Modulators in the Maturation of Cartilage to Prune Excessive Chondrocytes

This section illustrates the microscopic features of neocartilage 19 days post-graft implantation, highlighting the perichondrial papillae (PP), pre-resorptive layers (PRL), and vascular canals (VCs). The focus is on PP and PRL, utilizing H&E, safranin O/fast green staining, and immunohistochemistry (IHC) with αSMA antibodies, where “P” indicates perichondrium, and black arrows point to PRL (Figure 5A–C). It details the VCs, demonstrated through H&E, safranin O/fast green, alcian blue stains, and IHC with antibodies for type II collagen, Sox9, aggrecan, PCNA, αSMA, and CD31 (Figure 5D–L). The protein expression is shown in brown, while safranin O/fast green appears red, and alcian blue in blue. All images are magnified at 200×, providing a detailed view of the tissue’s staining characteristics and protein expression related to cartilage development and vascularization.

### 3.5. Concise Four-Stage Developmental Model to Elucidate the Intricate Process of Tracheal Cartilage Regeneration Based on the Occurrence of Key Chondrogenesis Features

This figure provides a comprehensive overview of the stages of chondrogenesis in graft neotissue, characterized by histological changes and specific protein expressions: It shows the progression of cartilage through five distinct stages, revealing changes in cell morphology and structure by H&E stain (Figure 6A–D). It indicates the presence of glycosaminoglycans (GAGs) in the cartilage matrix, with blue coloring signifying their distribution and density by alcian blue stain (Figure 6E–H). It displays the GAGs in red, with the vascular canal modulator structure identified by black arrows in safranin O/fast green stain (Figure 6I–L). It quantifies the number of chondrocytes and the ratios of protein expressions (Sox9, aggrecan, and PCNA) across different chondrogenesis stages, highlighting the dynamic nature of cartilage development within the neotissue. All images are magnified to show fine details, and the data presented include mean values with standard deviations, noting statistically significant differences where *p* < 0.05 (Figure 6M–P).

### 3.6. Types of Cartilage Evolution during Chondrogenesis

The figure presents a series of histological sections that illustrate the stages of cartilage development during chondrogenesis, as indicated by H&E and type II collagen stains. It shows early-stage chondrogenesis with tapering spear-like neocartilage exhibiting elongation within the tracheal structure, suggesting initial cartilage formation (Appendix A). This reflects the intermediary stage where separate cartilage templates are in the process of merging (Appendix A). It then demonstrates a more advanced stage where the cartilage templates have fused, with type II collagen expression indicated by the brown coloration (Appendix A). All images are magnified at 100× to detail the progressive structural changes and protein expression that occur as neocartilage matures.

## 4. Discussion

In our journey to advance tracheal transplantation techniques, nearly 40 extensive animal experiments have significantly contributed to our understanding of tracheal tissue regeneration. We harnessed 3D printing technologies to evolve from Crystal and Silicone to the biodegradable polycaprolactone (PCL), which proved instrumental in enhancing tissue ingrowth due to its design featuring simulated cartilaginous rings (Appendix A) [31]. This design not only ensured the necessary mechanical strength and flexibility but also paved the way for effective tissue integration. Despite initial experiments having had short survival spans, these trials were crucial in revealing early tissue regeneration dynamics and morphological transformations, casting light on aspects akin to embryonic organ regeneration. This underscores the benefits of leveraging large animal models to obtain comprehensive observations—a feat challenging to achieve with smaller animal models due to their inherent limitations [17,18,19,20].

A pivotal finding from our recent endeavors is the survival of two large animals beyond 90 days, illustrating complete regeneration of cartilage rings, mucous membranes, and additional phenomena such as polyp formation and ossification, typically associated with infections (Figure 1, Figure 2 and Figure 3). This significant results underscores the efficacy of biodegradable PCL grafts, fabricated through bioprinting for large animal applications, as potential viable clinical solutions [16]. Notably, the use of biodegradable PCL without tracheal stents in large animals represents a groundbreaking achievement.

Despite the biodegradability and tissue compatibility of PCL tracheal implants, a duration of approximately two to three years is required for their complete degradation to allow full tracheal tissue regeneration at the reconstruction site [6,15]. This prolonged process exposes the implant to environmental factors, which communicates with the environment at the upper tracheal end, is exposed to varying pH levels, humidity, impurities, and potential lung infections, potentially leading to complications such as infection, inflammation, and necrosis, which could compromise the integrity of the implant, resulting in tracheal collapse. To mitigate these risks, we adopted an innovative approach by utilizing reinforcement suture over distal trachea and strap muscle to release tension at the anastomosis site, thereby enhancing stability and resistance against the thoracic cavity’s negative pressure (Appendix A). Further, to address the issue of excessive granulation tissue, we applied laser therapy to ensure the airway remained unobstructed, enabling the animals’ survival beyond 90 days (Appendix A). This combination of innovative techniques has been crucial in our success, suggesting the future potential of enhancing tracheal implants with coatings similar to drug-eluting stents for reduced granulation tissue formation and improved postoperative outcomes [32,33]. Postoperatively, the administration of antibiotics via muscle injections could actively prevent infections and their severity, potentially extending the duration of survival even further.

Our observations upon implantation revealed two distinct cartilage tissue growth patterns: (1) the spear-like extension of chondroblasts in some regions, indicating initial cartilage formation; and (2) the fusion of cartilage templates in others, signifying advanced stages of cartilage development. This variability highlights the complex nature of cartilage regeneration within the neotissue, influenced by factors such as nutrient supply and waste removal facilitated by structures derived from the perichondrium. The roles of VC, PP, and PRL derived from the perichondrium are essential in supplying nutrients, metabolizing chondrocyte products, and eliminating metabolites through the submucosa and epithelium areas (as indicated by alcian blue staining and CD31 marker) [31]. Ninety-two days post-implantation, our examination of regenerated tissues offered profound insights, especially noting extensive ossification near the anastomotic ends. This suggests a link between tissue ossification and infection-induced inflammatory responses, offering valuable perspectives on the regeneration process.

Addressing postoperative care challenges, particularly in preventing complications like stenosis due to intraluminal granulation tissue, remains a priority. Our research underscores the promising potential of 3D-printed PCL scaffolds in tracheal tissue regeneration, showcasing improved survival and integration in large animal models. The surgical techniques and postoperative care strategies developed through this research, including the strategic use of strap muscle for graft stabilization and laser ablation for granulation tissue management, represent significant advancements in overcoming the hurdles associated with tracheal transplantation. Moreover, the prospective application of mTOR inhibitors on the scaffold surfaces opens new avenues for enhancing the success and longevity of tracheal implants, paving the way for novel approaches to tracheal repair and regeneration in clinical settings.

The limitations of this study include: (a) The sample size of pigs utilized in the research was still small, the need for a greater number of successes in future endeavors; (b) The postoperative care and management of large animal present significant challenges, particularly when administering anesthesia to individuals exceeding 100 kg. It is imperative to address and surmount these challenges. One potential solution could involve the utilization of mini-pigs; (c) In an effort to increase survival durations and diminish the likelihood of postoperative infections, it is essential to improve the animals’ living conditions. This entails the enhancement of sanitary practices and the regulation of environmental conditions, such as temperature and humidity, to support the well-being of the animals.

## 5. Conclusions

Our research substantiates the significant potential of 3D-printed PCL scaffolds in tracheal tissue regeneration, emphasizing the importance of innovative surgical techniques and postoperative care strategies in overcoming the challenges of tracheal transplantation. Our results have demonstrated that: (1) The improved surgical technique has successfully overcome the challenge of thoracic negative pressure, ensuring the stability of the implant and trachea; (2) After implantation, overcoming the dilemma of technique of general anesthesia, laser ablation technology is applied to manage granulation tissue, maintaining airway patency for long-term survival; (3) This stability has allowed the animals to survive for more than 90 days, proving the technique’s effectiveness. However, achieving complete tissue regeneration requires the full degradation of PCL, which takes two to three years. Therefore, our further experiment is to employ Lanyu mini-pigs for tracheal transplantation, effectively controlling weight gain to achieve the goal of successful and complete tissue regeneration.

## Figures and Tables

**Figure 1 bioengineering-11-00832-f001:**
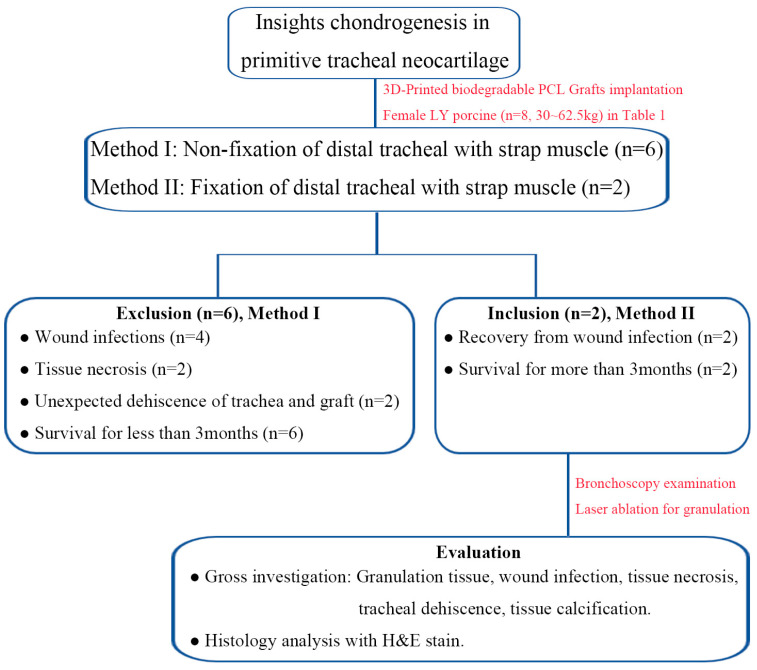
A schematic representation of the experimental design of the tracheal transplantation using 3D-printed biodegradable PCL grafts in female LY porcines. Method I, which did not involve fixation of the distal trachea, resulted in exclusions due to various complications. In contrast, Method II, which employed fixation, led to successful inclusions and prolonged survival. The evaluation process included bronchoscopy, laser ablation, gross investigation, and histological analysis with H&E staining to assess neocartilage development and address complications.

**Figure 2 bioengineering-11-00832-f002:**
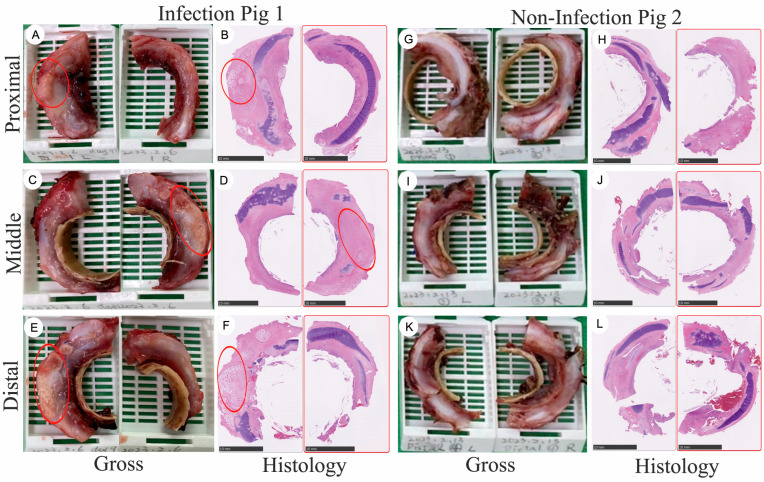
Neotissue growth outside the graft from the proximal to the distal region with gross and histology images following implantation into large-scale porcine models for over 90 days. This figure illustrates the growth of neotissue in two porcine subjects, one with a soft tissue infection and the other without, following the implantation of tracheal grafts for over 90 days. The histological analysis of the neotissue was conducted at the proximal (**A**,**B**), middle (**C**,**D**), and distal (**E**,**F**) regions of the infected Pig1. Red circles indicate areas of heterotopic ossification. Similarly, the histology of the neotissue was examined at the proximal (**G**,**H**), middle (**I**,**J**), and distal (**K**,**L**) regions of the non-infected Pig2. The gross images display intact tracheal rings, while the histology images, stained with H&E, highlight the development of cartilage and areas of heterotopic ossification, indicated by red circles. The neotissue extends from the proximal to the distal region outside the graft. Intact cylindrical neotissue, including cartilage, is visible in sections (**A**,**C**,**E**,**I**,**K**), with section (**G**) displaying only a half-ring of neotissue. The scale bars in the histology images represent 10 mm, providing a reference for the size of the observed features.

**Figure 3 bioengineering-11-00832-f003:**
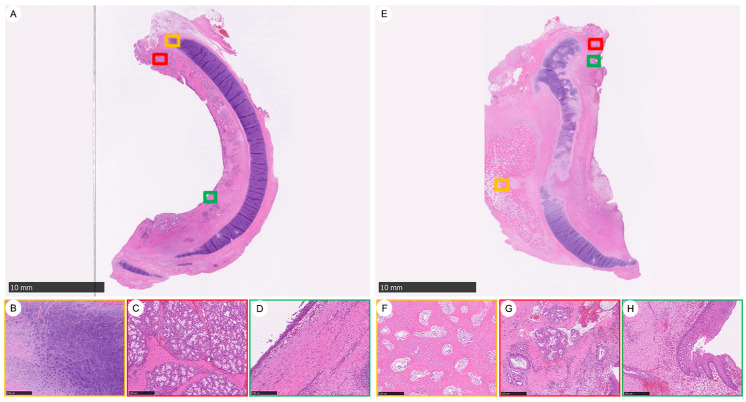
A Comparative Histological Analysis of Tracheal Tissue Regeneration in Two Large-Scale Porcine Models 90 Days Post-Implantation. (**A**–**D**) exhibit a regenerated tracheal section with a fully integrated structure, showcasing robust neocartilage formation at the tip (**B**), dense clusters of newly formed submucosal glands (**C**), and a smooth epithelial lining (**D**). The total view (**A**) highlights the well-organized tissue with clear chondrogenesis and glandular development. (**E**–**H**) display a contrasting tracheal section where irregularities are more pronounced. There is evident heterotopic ossification (**E**), with the central area (**F**) showing lamellar bone trabeculae amidst fibrotic-like marrow tissue. The submucosal glands are irregularly clustered (**G**), and the epithelium appears uneven (**H**). The total view (**E**) underscores the disparity in tissue organization compared to the left panels.

**Figure 4 bioengineering-11-00832-f004:**
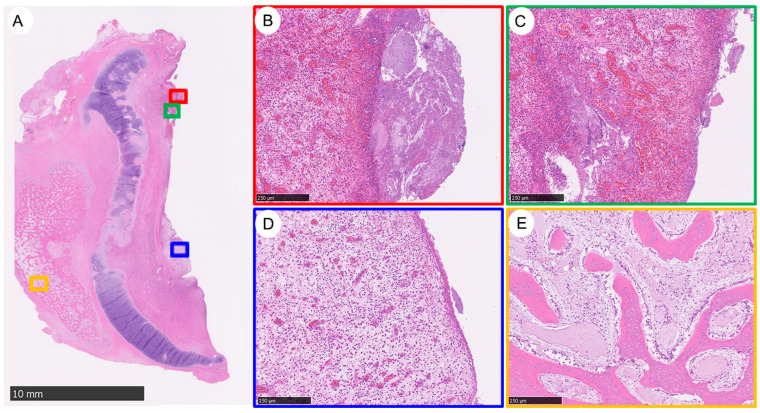
Histology at the Proximal Area of Neotissue Features Following Implantation into Large-Scale Porcine for Over 90 Days. (**A**) Heterotopic ossification; (**B**) Polyp; (**C**) Bacterial clump; (**D**) Congestion in the submucosal area. Yellow, red, blue, and green squares are magnified at each indicated image (**B**–**E**) at 100×.

**Figure 5 bioengineering-11-00832-f005:**
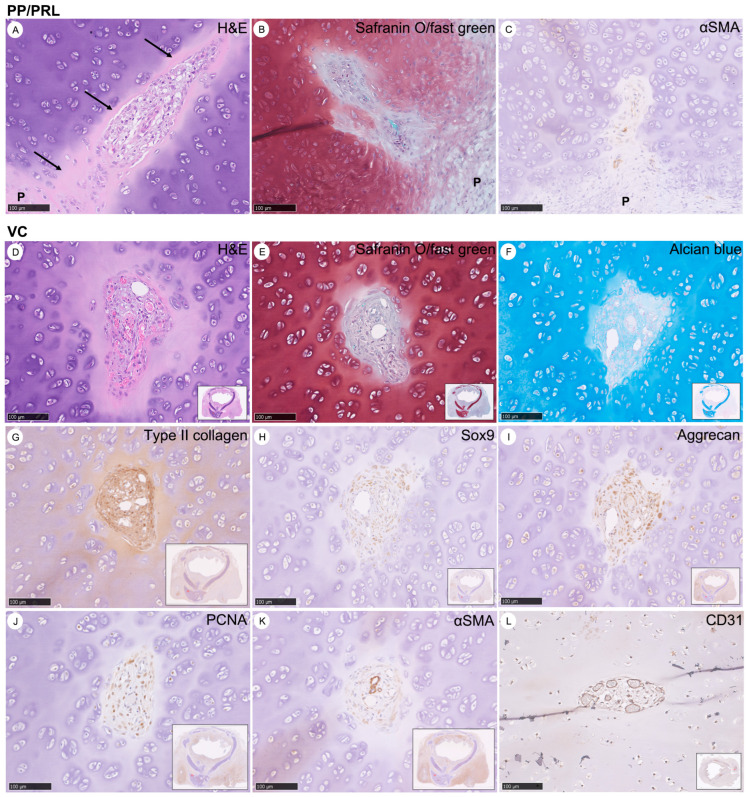
Features of Perichondrial Papillae (PP), Pre-resorptive Layers (PRL), and Various Stains of Vascular Canals (VCs) in Neocartilage after Graft Implantation for over 19 Days. (**A**–**C**) The PP and PRL were detected by H&E, safranin O/fast green, and IHC with αSMA antibodies. “P” is indicated in the perichondrium; black arrows represent PRL. (**D**–**L**) The VCs were detected by H&E, safranin O/fast green, alcian blue stains, and IHC with antibodies for type II collagen, Sox9, aggrecan, PCNA, αSMA, and CD31. Protein expression is indicated in brown; safranin O/fast green appears red; alcian blue is shown in blue. These data were magnified at 200×.

**Figure 6 bioengineering-11-00832-f006:**
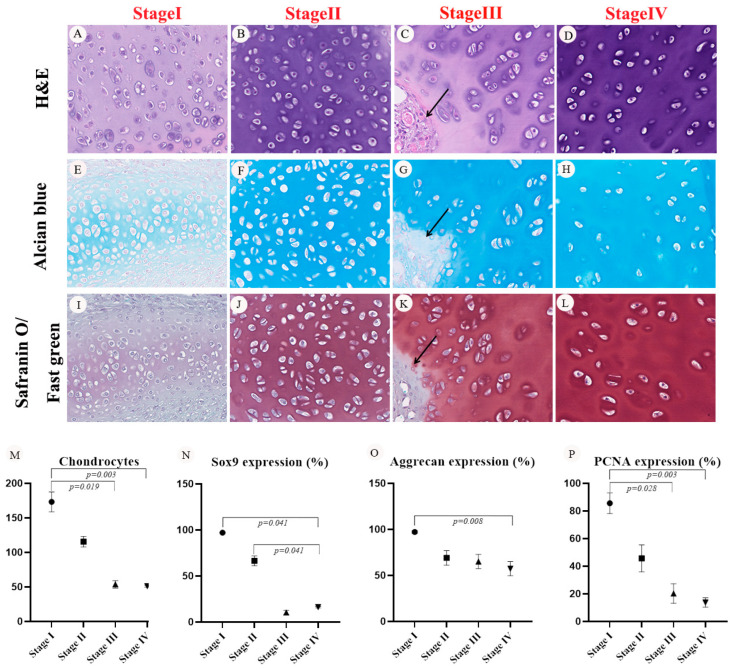
Proposed Four-Stage Chondrogenesis within the Graft Neotissue as Revealed by H&E, Alcian Blue, Safranin O/Fast Green, and IHC (Type II Collagen, Sox9, Aggrecan, and PCNA Antibodies) Stains. (**A**–**D**) Different stages of cartilage during chondrogenesis were investigated through H&E staining. (**E**–**H**) The histology and GAG content of the different stages of cartilage during chondrogenesis were investigated using Alcian blue stain. The GAGs of the cartilage matrix were detected by Alcian blue and are indicated in blue. (**I**–**L**) The histology and GAG content of the different stages of cartilage during chondrogenesis were investigated with safranin O/Fast Green stain. The GAGs of the cartilage matrix were detected with safranin O/Fast Green, indicating a red color. Black arrows: the modulator structure of the vascular canal (VC). Images were at 400× magnification. (**M**) Chondrocyte numbers quantified for different stages of chondrogenesis of graft neotissue using H&E stain (*n* = 4 views at 400× magnification for each group). (**N**–**P**) Ratios of protein expressions (Sox9, Aggrecan, and PCNA) quantified for different stages of chondrogenesis of graft neotissue based on IHC stainings (*n* = 4 views at 200× magnification for each group). Data are shown as mean ± SD values, and statistically significant differences across various stages were determined at *p* < 0.05.

**Table 1 bioengineering-11-00832-t001:** Demographics of PCL tracheal grafts transplanting into porcine model.

Item	Animals	
	Porcine 1	Porcine 2
Weight, kg	60	62.5
Number of bronchoscopy	9	10
Number of laser ablation for granulation tissue	8	7
Complication	Granulation Infection (soft tissue) Calcification	Granulation
Weight gain, kg	100	101
Survival day	92	92
Cause of death (no. of animal)	Sacrifice	Sacrifice

## Data Availability

The original contributions presented in the study are included in the Appendix A; further inquiries can be directed to the corresponding author.

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
