# Peer review of "Long-Term Survival and Regeneration Following Transplantation of 3D-Printed Biodegradable PCL Tracheal Grafts in Large-Scale Porcine Models"

_bioengineering, 2024, doi:10.3390/bioengineering11080832_

Round 1

Reviewer 1 Report

Comments and Suggestions for Authors

The submitted work discusses the outcomes of preclinical trials on trachea regeneration using a polycaprolactone implant in a pig model. The paper provides a detailed account of the procedure and the effects post-implantation, as evidenced by the tissue morphology surrounding the implant. While the study is compelling in its description of the process and its outcomes, the presented results are somewhat contentious. Publication of the manuscript will require additional information and clarifications. Only after sending the required clarifications and finalizing the content will this manuscript be sent for printing.

Below are questions that need to be addressed:

Provide detailed characteristics of the polymer used in forming the implants. Include information about the source of polycaprolactone (or the method and process of its synthesis), its number average molecular mass, and the dispersion of molecular weights.

Briefly outline the procedure for forming the implants, the final shape of the implants, and most importantly, provide an analysis of their surface and suitability for cell colonization. This analysis should include SEM microscopy, surface morphology, roughness measurements, water wetting angle, etc. The literature suggests that independent colonization of the surface of implants or scaffolds by mammalian cells is challenging (Journal of Orthopaedic Translation Volume 18, July 2019, Pages 128-141), especially when creating a regenerating endotracheal implant, which requires special preparation of its structure (https://doi.org/10.1002/hed.23343).

The main issue with the research is the unprofessional method used to sterilize the implants before the surgical procedure. The method involved immersing the implants in ethanol and exposing them to UV radiation, which may not guarantee complete sterility as required for this type of polymer implant. According to medical practice, this type of implant should be sterilized using ethylene oxide, gamma radiation, or fast electrons at a dose of 25KGy.

The uncertainty regarding the complete sterilization of the implants is linked to the uncertainty of the conclusions drawn about the observed post-implantation effects, particularly the occurrence of bacterial infection in some of the studied animals. The strong bacterial infection observed in the presented method may be generally attributed to inadequate sterilization. It's possible that similar results would be obtained for both methods with effective and certain sterilization, which was lacking in this case.

Unfortunately, the tests should be repeated using samples that have undergone standard sterilization, for example, by exposing the samples to electron radiation at a dose of 25 kGy.-

I would like to understand the tissue regeneration process on the implant surface and how tracheal cells colonize it. Since PCL is a highly hydrophobic material, it may limit cell settlement on its surface. We should investigate whether the implant surface contains caverns and cavities where tracheal cells can settle and grow. Additionally, we need to determine if bacteria create a biofilm that can be colonized by cells. We require an analysis of the formed implant surface, SEM microscope images, and measurements of surface smoothness. It's important to verify if the tracheal tissue regeneration process is controlled, or if it could lead to overgrowth of the tracheal lumen at a later stage. We also need to ensure that the thickness of the regenerated trachea section does not exceed the original thickness.

The study did not confirm whether complete tissue regeneration and full functionality transfer will occur in the long term (it takes about two years for the polycaprolactone implant to completely biodegrade). It is essential to validate these results over a longer period, preferably 2-3 years using the mini pig model. This must be clearly explained in the conclusions.

Comments on the Quality of English Language

Quite a few grammatical and stylistic errors

Author Response

Comments 1:

[(1) Provide detailed characteristics of the polymer used in forming the implants.

(2) Include information about the source of polycaprolactone (or the method and process of its synthesis), its number average molecular mass, and the dispersion of molecular weights.]

Response 1: Thank you for pointing this out. We agree with this comment. Therefore, we have revised it to state: (1) " The fabrication commenced with ester-terminated PCL (CAS-n 24980-41-4, CELLINK, TP-60505) pellets, a biodegradable polyester, …" Moreover, the PCL we used has varied molecular weights depending on the fabrication procedures, as there is no fixed molecular weight. This revised text can be found on page 3, first paragraph, lines 101-102.

(2) the method and process of PCL synthesis have added with “To summarize, we used Cellink HeartWare software to edit the G-code printing program. We produced the best graft by printing with both sterilized support material 'CELLINK START' (CELLINK, IK-190000) and PCL in different cartridges. This method comprised alternating PCL and washable support materials for the inner and outer layers, with PCL serving as the filler material. The maximum flexibility of the implant was attained with 20mm in length, 20 mm in diameter, a thickness of 1.5 mm and 25% infill. This method created implants with fine and regular pores, resulting in optimal printing conditions. The final PCL grafts featured four deliberately placed perforations to promote uniform cell ingrowth and integration with the native trachea after implantation (S2Figure B).” This revised text can be found on page 3, paragraph 1, lines 103-112.

Comments 2:

[(1) Briefly outline the procedure for forming the implants, the final shape of the implants, and most importantly, provide an analysis of their surface and suitability for cell colonization. This analysis should include SEM microscopy, surface morphology, roughness measurements, water wetting angle, etc.

(2) The literature suggests that independent colonization of the surface of implants or scaffolds by mammalian cells is challenging (Journal of Orthopaedic Translation Volume 18, July 2019, Pages 128-141), especially when creating a regenerating endotracheal implant, which requires special preparation of its structure (https://doi.org/10.1002/hed.23343).]

Response 2: (1) Agree. We have revised the procedure for forming the implants in the Materials and Methods section. The revised text now reads: ” ”The fabrication commenced with ester-terminated PCL (CAS-n 24980-41-4, CELLINK, TP-60505) pellets, a biodegradable polyester, heated to a molten state, then extruded in layers to form the graft, requiring 3-4 hours. To summarize, we used Cellink HeartWare software to edit the G-code printing program. We produced the best graft by printing with both sterilized support material 'CELLINK START' (CELLINK, IK-190000) and PCL in different cartridges. This method comprised alternating PCL and washable support materials for the inner and outer layers, with PCL serving as the filler material. The maximum flexibility of the implant was attained with 20mm in length, 20 mm in diameter, a thickness of 1.5 mm and 25% infill. This method created implants with fine and regular pores, resulting in optimal printing conditions. The final PCL grafts featured four deliberately placed perforations to promote uniform cell ingrowth and integration with the native trachea after implantation (S2Figure B).” This revised text can be found on page 3, paragraph 1, lines 101-112.

  However, according to current research literature, there are only results from studies involving the implantation of PCL in small animals, and the survival time in large animal experiments has not exceeded three weeks. Therefore, our research plan aims to design 3D-printed PCL implants for implantation in large animals for more than 90 days to observe tissue regeneration in vivo. Given that our research environment requires collaboration with high-level research institutions to perform electron microscope analysis of the implants, the process is cumbersome and complex. This will be the focus of our next advanced experimental plan. Additionally, the regenerated tissues are embedded in wax blocks, and during the process of making these wax blocks, the PCL implants dissolve. This makes it impossible to accurately observe whether cells are attaching and growing on the implants, and we can only observe new tissue formation around the implants. In the future, we plan to collaborate with experts to conduct an analysis of the formed implant surface, including SEM microscope images and measurements of surface smoothness. We aim for the implants to achieve at least six months of survival in vivo, which would be more beneficial for clinical applications.

(2) We agree with the findings presented in "Journal of Orthopaedic Translation Volume 18, July 2019, Pages 128-141," which indicate that the current in vivo experiments using Pure-PCL implants facilitate the homing of stem cells within the body. Previous studies have successfully isolated and cultured mesenchymal stem cells (MSCs) from the Wharton's jelly of umbilical cords in vitro. In future research, we plan to incorporate these stem cells into the implants to create biomimetic tracheal implants, aiming to improve implantation outcomes.

Comments 3:

[(1) The main issue with the research is the unprofessional method used to sterilize the implants before the surgical procedure. The method involved immersing the implants in ethanol and exposing them to UV radiation, which may not guarantee complete sterility as required for this type of polymer implant. According to medical practice, this type of implant should be sterilized using ethylene oxide, gamma radiation, or fast electrons at a dose of 25KGy.

(2) The uncertainty regarding the complete sterilization of the implants is linked to the uncertainty of the conclusions drawn about the observed post-implantation effects, particularly the occurrence of bacterial infection in some of the studied animals. The strong bacterial infection observed in the presented method may be generally attributed to inadequate sterilization. It's possible that similar results would be obtained for both methods with effective and certain sterilization, which was lacking in this case. Unfortunately, the tests should be repeated using samples that have undergone standard sterilization, for example, by exposing the samples to electron radiation at a dose of 25 kGy.- ]

Response 3:

(1) Thank you for pointing this out. We agree with this comment. Considering that implantation in animals may easily cause infections, and as a thoracic surgeon myself, I am very concerned about the issue of insufficient sterilization of implants. However, due to the melting point of PCL being 60℃, it is not possible to use the commonly employed high-temperature and high-pressure sterilization methods in the laboratory. After reviewing relevant literature on implant sterilization and applying the sterilization method described in the paper by Kim et al., 2019, to our tracheal implants, we confirmed that there was no contamination through in vitro culture testing before proceeding with animal trials. We have cited the reference "Kim et al., 2019" on page 3, first paragraph, lines 116.

(2) The surgical procedures and postoperative care for large animals differ significantly from those for humans, presenting considerable challenges for long-term survival. Unlike humans, animals cannot cooperate with changing dressings or receiving intravenous antibiotics and pain medications. Additionally, the environment of a pigsty cannot be compared to a clinical ward, which further complicates postoperative care. To address these issues, we will closely collaborate with the veterinary team to develop strategies for improving postoperative infection management. This collaboration will focus on enhancing the sanitary conditions, optimizing medication delivery methods, and ensuring more effective monitoring and care of the animals post-surgery.

Comments 4:

[(1) I would like to understand the tissue regeneration process on the implant surface and how tracheal cells colonize it. Since PCL is a highly hydrophobic material, it may limit cell settlement on its surface. We should investigate whether the implant surface contains caverns and cavities where tracheal cells can settle and grow. Additionally, we need to determine if bacteria create a biofilm that can be colonized by cells.

(2) We require an analysis of the formed implant surface, SEM microscope images, and measurements of surface smoothness. It's important to verify if the tracheal tissue regeneration process is controlled, or if it could lead to overgrowth of the tracheal lumen at a later stage. We also need to ensure that the thickness of the regenerated trachea section does not exceed the original thickness.]

Response 4:

(1) Thank you for the knowledgeable suggestion. The formation of biofilm is indeed very likely because the implant surface is designed with pores and gaps, making it easy for mucus to accumulate and for bacteria to colonize. Therefore, the dosage and duration of postoperative antibiotics are crucial.

(2) Additionally, the regenerated tissues are embedded in wax blocks, and during the process of making these wax blocks, the PCL implants dissolve. This makes it impossible to accurately observe whether cells are attaching and growing on the implants, and we can only observe new tissue formation around the implants. In the future, we plan to collaborate with experts to conduct an analysis of the formed implant surface, including SEM microscope images and measurements of surface smoothness. We also aim to collect regenerated tracheal tissue samples from long-term animal studies (with implants in place for at least six months). Some of these tissues will be analyzed using cryosectioning to observe the growth and regeneration of cells on the PCL implants. Furthermore, friction at the junction between the implant and the trachea can lead to the formation of granulation tissue, causing excessive tissue proliferation.

Comments 5: [The study did not confirm whether complete tissue regeneration and full functionality transfer will occur in the long term (it takes about two years for the polycaprolactone implant to completely biodegrade). It is essential to validate these results over a longer period, preferably 2-3 years using the mini pig model. This must be clearly explained in the conclusions.]

Response 5: Thank you for the insightful comment. We have revised the conclusion with “Our results have demonstrated that (1) the improved surgical technique has successfully overcome the challenge of thoracic negative pressure, ensuring the stability of the implant and trachea. (2) After implantation, overcoming the dilemma of technique of general anesthesia, laser ablation technology is applied to manage granulation tissue, maintaining airway patency for long-term survival. (3) This stability has allowed the animals to survive for more than 90 days, proving the technique's effectiveness. However, achieving complete tissue regeneration requires the full degradation of PCL, which takes 2-3 years. Therefore, our further experiment is to employ Lanyu mini-pigs for tracheal transplantation. By effectively controlling weight gain to achieve the goal of successful and complete tissue regeneration.” This change can be found in the revised manuscript on page 13, paragraph 4, lines 455-464.

Reviewer 2 Report

Comments and Suggestions for Authors

General comments

The submitted manuscript consists in a research paper about the application of 3D-printed, biodegradable PCL-based tracheal grafts for large-scale porcine tracheal transplantation.

The paper topic is very interesting and worthy of investigation. More expertimental details have to be added.

Detailed comments and remarks are listed below point by point.

Abstract

-        Please add a contextualization.

-        All the acronyms, such as PCL; have to be specified the first time they are used.

-         

-        1. Introduction
- Please, better justify the use of additive manufacturing technologies for biomedical applications, using recent references, including “Bioprinting technology in skin, heart, pancreas and cartilage tissues: Progress and challenges in clinical practice. International journal of environmental research and public health, 18(20)(2021), 108060” and “3D bioprinting in airway reconstructive surgery: A pilot study. International Journal of Pediatric Otorhinolaryngology, 161 (2022), 111253.”.
- The sentences “PCL, known for its mechanical strength and biocompatibility, stands out among biodegradable polymers for its suitability in 3D printing applications. This eliminates the need for harmful solvents, allowing for the fabrication of scaffolds that closely replicate the intricate architecture of the natural trachea.” have to be supported with proper references.

-         

2. Materials and Methods

2.1. Design and Preparation of PCL Tracheal Grafts

- More details about the used PCL have to be added, such as the supplier and the molecular weight.

- More details about the design and the sample dimensions have to be added.

- More details about the printing parameters have to be added.

3. Results

Please, add the main results about the printed samples, commeting the images reported in the supplementary within the text body.

5. Conclusions

The Conclusions section has to be improved and extended, adding some numerical data and the main achievements.

Comments on the Quality of English Language

English is good

Author Response

Comments 1:

Abstract

-        (1) Please add a contextualization.

-        (2) All the acronyms, such as PCL; have to be specified the first time they are used.

Response 1:

(1) Thank you for your excellent suggestion. We have added a contextualization in abstract and introduction with “The polycaprolactone (PCL) implants in large animals shows great promise for tracheal transplantation. However, the longest survival time achieved to date is only about three weeks. To meet clinical application standards, it is essential to extend the survival time and ensure the complete integration and functionality of the implant.”

(2) Thank you for your correction. We have revised all the acronyms in the abstract. This change can be found on page 1, paragraph 1, line 27-30.

Comments 2: [1. Introduction
- Please, better justify the use of additive manufacturing technologies for biomedical applications, using recent references, including “Bioprinting technology in skin, heart, pancreas and cartilage tissues: Progress and challenges in clinical practice. International journal of environmental research and public health, 18(20)(2021), 108060” and “3D bioprinting in airway reconstructive surgery: A pilot study. International Journal of Pediatric Otorhinolaryngology, 161 (2022), 111253.”.
- The sentences “PCL, known for its mechanical strength and biocompatibility, stands out among biodegradable polymers for its suitability in 3D printing applications. This eliminates the need for harmful solvents, allowing for the fabrication of scaffolds that closely replicate the intricate architecture of the natural trachea.” have to be supported with proper references.
]

Response 2: We agree with this comment. We have cited the relevant literature as references 10-11 to support the statement: “PCL, known for its mechanical strength and biocompatibility, stands out among biodegradable polymers for its suitability in 3D printing applications.” Additionally, references 12-14 were cited to support the statement: “This eliminates the need for a time-consuming decellularization process and harmful solvents.” We have accordingly modified this section to emphasize these points. This change can be found on page 2, paragraph 1, lines 59-60.

Reference10: Wang, C.;  Dong, J.;  Liu, F.;  Liu, N.; Li, L., 3D-printed PCL@BG scaffold integrated with SDF-1α-loaded hydrogel for enhancing local treatment of bone defects. J Biol Eng 2024, 18 (1), 1.

Reference11: Huber, F.;  Vollmer, D.;  Vinke, J.;  Riedel, B.;  Zankovic, S.;  Schmal, H.; Seidenstuecker, M., Influence of 3D Printing Parameters on the Mechanical Stability of PCL Scaffolds and the Proliferation Behavior of Bone Cells. Materials (Basel) 2022, 15 (6).

Reference12: de Wit, R. J. J.;  van Dis, D. J.;  Bertrand, M. E.;  Tiemessen, D.;  Siddiqi, S.;  Oosterwijk, E.; Verhagen, A., Scaffold-based tissue engineering: Supercritical carbon dioxide as an alternative method for decellularization and sterilization of dense materials. Acta Biomater 2023, 155, 323-332.

Reference13: Guimaraes, A. B.;  Correa, A. T.;  Pego-Fernandes, P. M.;  Maizato, M. J.;  Cestari, I. A.; Cardoso, P. F., Biomechanical Properties of the Porcine Trachea before and after Decellularization for Airway Transplantation. The Journal of Heart and Lung Transplantation 2021, 40 (4, Supplement), S385-S386.

Reference14: V, G. R.;  Wilson, J.;  L, V. T.; Nair, P. D., Assessing the 3D Printability of an Elastomeric Poly(caprolactone-co-lactide) Copolymer as a Potential Material for 3D Printing Tracheal Scaffolds. ACS Omega 2022, 7 (8), 7002-7011.

Comments 3: [2. Materials and Methods

2.1. Design and Preparation of PCL Tracheal Grafts

- More details about the used PCL have to be added, such as the supplier and the molecular weight.

- More details about the design and the sample dimensions have to be added.

- More details about the printing parameters have to be added.]

Response 3: Agree. We have revised the procedure for forming the implants in the Materials and Methods section. The revised text now reads: ”The fabrication commenced with ester-terminated PCL (CAS-n 24980-41-4, CELLINK, TP-60505) pellets, a biodegradable polyester, heated to a molten state, then extruded in layers to form the graft, requiring 3-4 hours. To summarize, we used Cellink HeartWare software to edit the G-code printing program. We produced the best graft by printing with both sterilized support material 'CELLINK START' (CELLINK, IK-190000) and PCL in different cartridges. This method comprised alternating PCL and washable support materials for the inner and outer layers, with PCL serving as the filler material. The maximum flexibility of the implant was attained with 20mm in length, 20 mm in diameter, a thickness of 1.5 mm and 25% infill. This method created implants with fine and regular pores, resulting in optimal printing conditions. The final PCL grafts featured four deliberately placed perforations to promote uniform cell ingrowth and integration with the native trachea after implantation (S2Figure B).” This revised text can be found on page 2, first paragraph, lines 101-112.

Comments 4: [3. Results

Please, add the main results about the printed samples, commeting the images reported in the supplementary within the text body.]

Response 4: Thank you for the comment. We have modified content to Results with “The implanted PCL tracheal grafts with dimension of 20mm in length, 20 mm in diameter, a thickness of 1.5 mm and 25% infill (S2Figure B). “This revised text can be found on page 5, paragraph 2, lines 206-207.

Comments 5: [5. Conclusions

The Conclusions section has to be improved and extended, adding some numerical data and the main achievements.]

Response 5: Thank you for the insightful comment. We have revised the conclusion with “Our results have demonstrated that (1) the improved surgical technique has successfully overcome the challenge of thoracic negative pressure, ensuring the stability of the implant and trachea. (2) After implantation, overcoming the dilemma of technique of general anesthesia, laser ablation technology is applied to manage granulation tissue, maintaining airway patency for long-term survival. (3) This stability has allowed the animals to survive for more than 90 days, proving the technique's effectiveness. However, achieving complete tissue regeneration requires the full degradation of PCL, which takes 2-3 years. Therefore, our further experiment is to employ Lanyu mini-pigs for tracheal transplantation. By effectively controlling weight gain to achieve the goal of successful and complete tissue regeneration.” This change can be found in the revised manuscript on page 13, paragraph 4, lines 455-464.

Round 2

Reviewer 1 Report

Comments and Suggestions for Authors

The vast majority of my comments and requests for supplementation of the text were taken into account in the manuscript's revised text and the responses. I believe that the manuscript in its current form can be forwarded to further publishing stages. The work is exciting due to its practical significance and I wish the authors to introduce the described surgical technique into clinical practice.

Reviewer 2 Report

Comments and Suggestions for Authors

The paper can be accepted in the current revised version.

Comments on the Quality of English Language

The English is good.